# Lifetime cigarette smoking and chronic widespread and regional pain in later adulthood: evidence from the 1946 British birth cohort study

Rebecca Bendayan,[1,2] Rachel Cooper,[1] Stella G Muthuri[1]

[1]MRC Unit for Lifelong Health and Ageing at UCL, London, UK
[2]Department of Biostatistics and Health Informatics, Institute of Psychiatry, Psychology & Neuroscience (IoPPN), King's College London, London, UK

**Correspondence to**
Dr Rebecca Bendayan;
rebecca.bendayan@kcl.ac.uk

## ABSTRACT

**Objective** To examine whether different lifetime patterns of cigarette smoking are associated with chronic widespread pain (CWP) and chronic regional pain (CRP) at age 68.

**Design** Prospective cohort study.

**Setting** England, Scotland and Wales.

**Participants** Up to 2347 men and women from the Medical Research Council National Survey of Health and Development, who have been followed up since birth in 1946 and provided sufficient information on cigarette smoking across adulthood to be classified as never smoker, predominantly non-smoker, predominantly smoker or lifelong smoker and pain assessment at age 68.

**Primary outcome measures** Pain was self-reported at age 68, and CWP was defined according to American College of Rheumatology criteria. Participants who reported having pain for ≥3 months but who did not meet the CWP definition were classified as having CRP; those who reported pain which had lasted for <3 months were classified as 'other' pain. No pain was the reference group.

**Results** Findings from multinomial logistic regression models indicated that compared with never smokers, predominantly non-smokers, predominantly smokers and lifelong smokers all had an increased risk of CWP; relative risk ratios=1.70(95% CI 1.16 to 2.49); 2.10(95% CI 1.34 to 3.28) and 1.88(95% CI 0.99 to 3.57), respectively, after adjusting for sex, own occupational class, educational level, body mass index, leisure time physical activity, alcohol intake, long-standing illness and symptoms of anxiety and depression. No association was observed between smoking history and CRP or other pain.

**Conclusions** These results suggest that exposure to cigarette smoking at any stage in adulthood was associated with higher risk of CWP in later adulthood; highlighting the ongoing importance of smoking prevention programmes. It also suggests that assessment of lifetime smoking behaviour may be more useful in identifying those at greater risk of CWP in later life than assessment of current smoking status.

## Strengths and limitations of this study

► This study uses prospective data on smoking behaviour ascertained across adulthood (from age 20 to 68) providing us with the opportunity to characterise different lifetime patterns of smoking.
► This study distinguishes chronic widespread pain from regional pain and examines the risk of both.
► Chronic pain was assessed at a single time point so we were not able to establish temporality.
► Lifelong smokers were under-represented in the study sample as they had greater rates of all-cause mortality.
► Despite losses to follow-up over the 69 years of the study, the sample has remained broadly representative of the national population of England, Scotland and Wales born at a similar time.

## INTRODUCTION

Chronic pain is highly prevalent in older populations and has major adverse impacts on quality of life, well-being and activity participation.[1–4] Identifying modifiable risk factors associated with pain may inform interventions to prevent and/or alleviate its consequences in later life. Cigarette smoking is one of many modifiable behavioural risk factors that may play a role in the development and progression of chronic pain. While this risk factor has already been widely studied[5 6] with most longitudinal studies reporting weak or null associations,[7–10] the majority of these have been in populations younger than 65.

Moreover, even when studies have included older adults, these have usually only considered a single assessment of smoking status (either at baseline or at the current or most recent time point). This information has then been used to classify individuals as current, former or never smokers, or as ever or never smokers. This is unlikely to adequately capture the variability in lifetime exposure to smoking and its associated health risks due to changes in smoking behaviour across adulthood. Distinguishing people with different lifetime smoking histories and investigating differences in their health prospects may be especially important in older adults; although the prevalence of smoking

BMJ

has steadily declined across all age groups over the last few decades, the baby boom generation who are currently reaching old age were at high risk of exposure to smoking earlier in adulthood.[11] In addition, they are likely to have been exposed to antismoking campaigns and comprehensive tobacco control policies early enough in adulthood to have potentially benefited from the subsequent reductions in risk of smoking-related diseases linked to premature mortality.[12] While people with some history of smoking are therefore now more likely to reach old age than they would have been in previous generations, whether they reach old age with more chronic conditions and symptoms such as pain remains to be established.

Using data from the Medical Research Council (MRC) National Survey of Health and Development (NSHD), which has assessed smoking status at multiple time points across adulthood, this paper aims to examine whether different lifetime patterns of cigarette smoking are associated with chronic widespread pain (CWP) and chronic regional pain (CRP) at age 68.

## METHODS

### Study population

The MRC NSHD is a socially stratified population sample of 5362 single, legitimate births that occurred in England, Wales and Scotland in 1 week of March 1946. Participants have been prospectively followed from birth, and participation rates have remained high.[13–16] At the 24th data collection, conducted between 2014 and 2015 when study members were aged 68–69, study participants were first asked to complete a postal questionnaire at age 68 before then being invited to have a home visit by a research nurse at age 69.[15] Of the 2816 people in the target sample living in England, Scotland and Wales, 2370 (84.2%) completed a postal questionnaire. In addition, a postal questionnaire was sent to 126 study members living abroad who remain in contact with the study of whom 83 (65.9%) returned a questionnaire. No attempt was made to contact the remaining 2420 study members: 957 (17.8%) had already died, 620 (11.6%) had previously withdrawn from the study, 448 (8.3%) had emigrated and were no longer in contact with the study and 395 (7.4%) had been untraceable for more than 5 years.

### Patient and public involvement statement

Participants have a lifelong association with NSHD. Over the 72 years of the study, the research team has increasingly involved participants, in line with changing norms about conducting cohort studies, starting at age 16 (in 1962) with the annual dissemination of study findings in birthday cards. Participants receive personal letters whenever they raise queries or provide additional comments, including suggestions for new topics to study. In the last 10 years, the research team has increased the level of participant involvement through invitations to study events (65th and 70th birthday celebrations), public engagement activities and focus groups to discuss clinical

substudies. When piloting new questionnaires and assessments, we recruit patients from general practitioner practices or from the University College London Hospital patient public involvement group and take into account their feedback when designing the mainstage fieldwork.

### Assessment of pain

Information on pain was ascertained in the postal questionnaire administered at age 68. Participants were asked whether they had experienced any ache or pain in the last month which had lasted for 1 day or longer, not including pain occurring during the course of a feverish illness such as influenza. Those who reported pain were then asked to report whether or not they had experienced this pain for at least 3 months and to shade the location of their pain using a four-view body manikin. Using American College of Rheumatology criteria,[17] CWP was defined as pain present for 3 months or longer, both above and below the waist, on both the left and right side of the body and in the axial skeleton. Participants who reported having pain for at least 3 months but who did not meet the CWP definition were classified as having CRP and those who reported pain that had lasted for less than 3 months were classified as having 'other' pain.[18]

### Assessment of smoking

Data on cigarette smoking were obtained on eight occasions across adulthood using postal questionnaires (at ages 20, 25, 31, 60–64, 68–69) and nurse interviews (at ages 36, 43, 53). At each of these ages, study members were asked whether they currently smoked cigarettes. Participants who provided a negative response were asked whether they had ever smoked. This information was used to construct a set of variables indicating smoking status at each age from 20 to 68, coded as current, former and never smoker. Using these three-category variables, a lifetime smoking history variable was then constructed according to Clennell et al.[19] Using this approach, we included those who provided smoking data for at least three waves and for whom missing data were not sequential at three or more waves (n=3705). Based on their reported smoking status at each age across adulthood, participants were grouped into one of four categories to summarise their smoking history: (1) participants who reported being non-smokers at all available data collections were classified as 'never smokers'; (2) participants who reported being smokers at all available data collections were classified as 'lifelong smokers'; participants who were classified as smokers at some ages and as ex or never smokers at other ages were classified as: (3) 'predominantly non-smoker' if they were a non-smoker on at least three occasions and more often reported being a non-smoker than a smoker; (4) 'predominantly smoker' if they reported being a smoker on four or more occasions and more often reported being a smoker than a non-smoker. For example, if a study member reported being a smoker on five occasions and an ex-smoker on two occasions, they were classified as 'predominantly

smoker' whereas if they reported the reverse they were classified as 'predominantly non-smoker'.[19]

## Covariates

Potential confounders were selected a priori based on previous research on chronic pain and health behaviours.[20–23] These were sex, socioeconomic position (indicated by own occupational class (which also allows us to distinguish between manual and non-manual work) and educational level), body mass index (BMI), other health behaviours (leisure time physical activity and alcohol intake), long-standing illness and symptoms of anxiety and depression. Unless otherwise specified these were assessed at age 68. Own occupation at age 53 years (or if not available, the most recent measure in adulthood) was categorised according to the Registrar General's social classification into three groups: high (I (Professional) or II (Managerial and technical)); middle (IIINM (skilled non-manual (NM)) or IIIM (skilled manual (M))); low (IV (partly skilled manual) or V (unskilled manual)). Highest educational level achieved by age 26 was categorised into three groups: no qualifications; up to O-level or equivalent; A-level or equivalent and above. BMI ($kg/m^2$) was calculated from height (m) and weight (kg) measured by a trained nurse during the clinical assessment at age 69. Physical activity was assessed through self-reports of the level of participation in sports, vigorous leisure activities or exercise grouped as: inactive, moderately active (1–4 times/month) or most active (5 times or more/month). Similarly, alcohol intake in the last year was self-reported as never, only on special occasions, monthly or less, 2–4 times per month, 2–3 times per week, or 4 or more times per week. In addition, participants were asked whether they had any long-standing illness or health problems which had lasted, or were expected to last for 6 months or more. Symptoms of anxiety and depression were assessed using the 28-item General Health Questionnaire (GHQ-28). Each GHQ-28 item was coded using the General Scoring Method (ie, 0 for response choices 1 and 2; and as 1 or response choices 3 and 4). Symptoms of anxiety and depression were indicated by summing responses to all items and applying a threshold for caseness of 5 or more.[24]

## Statistical analyses

Multinomial logistic regression models were used to examine the associations between lifetime smoking history and pain outcomes (with no pain as the reference category). Models were first adjusted for sex and formal tests of interaction between sex and lifetime smoking history were undertaken. As no evidence of interactions with sex was found, subsequent models were sex adjusted. We then adjusted for each set of covariates in turn before adjusting for all covariates simultaneously in a final model. All models included the maximum number of participants with complete data on lifetime smoking history and pain (n=2347). To reduce data loss, covariates with missing values (ie, educational attainment

(n=114), occupational class (n=6), BMI (n=435), leisure time physical activity (n=25), alcohol intake (n=25), long-standing illness or health problems (n=24) and GHQ-28 (n=422) were imputed using multiple imputation by chained equations.[25] Analyses were performed across 20 imputed datasets and combined using Rubin's rules. In sensitivity analyses, models were rerun on the sample with complete data on all covariates (n=1759). All analyses were conducted using STATA V.14.1.

## RESULTS

Of the total analytical sample of 2347, 1304 (55.6%) reported pain, with 10.6% reporting CWP, 30.3% reporting CRP and 14.7% reporting 'other pain'. More women than men reported CWP (13.2% for women vs 7.7% for men, p<0.001). Overall, the prevalence of smoking declined across adulthood from 55.8% for men and 46.8% of women at age 20 to 9.9% for men and 7.8% for women by age 68 (p values for sex differences at both ages <0.001) (online supplementary table 1). The prevalence of lifelong non-smokers was 34.3% for women and 26.3% for men. Of those classified as predominantly non-smokers, 61.6% had stopped smoking at least once by age 25 and only 0.5% were smokers at the most recent report. Of those classified as predominantly smokers, 88.9% were smokers at age 25, but only 14.2% were still smokers at the most recent report. Descriptive statistics are shown in table 1. All covariates were associated with pain outcomes in the expected directions.

In sex-adjusted models, predominantly non-smokers, predominantly smokers and lifelong smokers all had a higher likelihood of CWP, compared with lifelong non-smokers (relative risk ratios (RRRs) of CWP vs no pain=1.80 (95% CI 1.26 to 2.58), 2.64 (95% CI 1.74 to 3.99) and 2.27 (95% CI 1.25 to 4.13), respectively (table 2, model 1). Adjustment for each set of covariates had only limited impact and associations were maintained in fully adjusted models; RRRs 1.70 (95% CI 1.16 to 2.49) for predominantly non-smokers, 2.10 (95% CI 1.34 to 3.28) for predominantly smokers and 1.88 (95% CI 0.99 to 3.57) for lifelong smokers (table 2, model 6). No associations were observed between lifetime smoking history and CRP or other pain (table 2).

When analyses were restricted to those with complete data (n=1759), we found similar results and the overall conclusions remained the same (online supplementary table 2).

## DISCUSSION

The main aim of this study was to examine whether different patterns of cigarette smoking over the life course were associated with CWP and CRP at age 68 in a nationally representative birth cohort. Our results showed that exposure to cigarette smoking at any time point across adulthood was associated with increased risk of CWP at age 68, even after adjustment for a range of

**Table 1** Characteristics of participants from the MRC National Survey of Health and Development by sex at age 68 (maximum n=2347)

| | Men, N* (%) | Women, N* (%) | P values† |
|---|---|---|---|
| Total | 1123 (47.9) | 1224 (52.1) | |
| **Pain at age 68** | | | |
| No pain | 537 (47.8) | 506 (41.3) | <0.001 |
| Other pain | 184 (16.4) | 162 (13.2) | |
| Chronic regional pain | 316 (28.1) | 394 (32.2) | |
| Chronic widespread pain | 86 (7.7) | 162 (13.2) | |
| **Lifetime smoking history** | | | |
| Never smoker | 295 (26.3) | 420 (34.3) | <0.001 |
| Predominantly non-smoker | 527 (46.9) | 516 (42.2) | |
| Predominantly smoker | 228 (20.3) | 224 (18.3) | |
| Lifelong smoker | 73 (6.5) | 64 (5.2) | |
| **BMI (kg/m² ) at age 69 (n=1919)** | | | |
| Mean (SD) | 28.0 (4.46) | 28.1 (5.56) | 0.92 |
| **Occupational class at age 53** | | | |
| High (I/II) | 631 (56.3) | 458 (37.5) | <0.001 |
| Middle (IIINM/IIIM) | 385 (34.4) | 543 (44.5) | |
| Low (IV/V) | 104 (9.3) | 220 (18.0) | |
| **Educational attainment by age 26** | | | |
| A-level or equivalent and above | 505 (47.2) | 374 (32.2) | <0.001 |
| Up to O-level or equivalent | 212 (19.8) | 419 (36.1) | |
| None | 354 (33.1) | 369 (31.8) | |
| **Alcohol intake at age 68** | | | |
| Never | 205 (18.5) | 396 (32.7) | <0.001 |
| Monthly or less | 55 (5.0) | 107 (8.8) | |
| 2–4 times per month | 157 (14.1) | 203 (16.8) | |
| 2–3 times per week | 295 (26.6) | 262 (21.6) | |
| ≥4 times per week | 398 (35.9) | 244 (20.1) | |
| **Participation in leisure time physical activity at age 68** | | | |
| Most active (≥5 times/month) | 319 (28.7) | 320 (26.4) | 0.18 |
| Moderately active (1–4 times/month) | 123 (11.1) | 161 (13.3) | |
| Inactive | 669 (60.2) | 730 (60.3) | |
| **Long-standing illness or health problem at age 68** | | | |
| No | 449 (40.3) | 508 (42.0) | 0.38 |
| Yes | 666 (59.7) | 700 (57.9) | |

Continued

**Table 1** Continued

| | Men, N* (%) | Women, N* (%) | P values† |
|---|---|---|---|
| **Symptoms of anxiety and depression (GHQ-28) at age 68** | | | |
| No (≤4) | 840 (90.7) | 829 (83.0) | <0.001 |
| Yes (>4) | 86 (9.3) | 170 (17.0) | |

*Sample restricted to those participants with data on smoking history and pain. N varies due to missing data on covariates.
†Comparison of sexes using Student's t-test or χ² tests as appropriate.
BMI, body mass index; GHQ-28, 28-item General Health Questionnaire; MRC, Medical Research Council.

potential confounders and mediators. However, there was no evidence of associations between lifetime cigarette smoking and CRP or other pain.

### Comparisons with other studies

Our finding of an association between different patterns of lifetime smoking and CWP in older adults builds on findings from other previous studies of chronic pain in older adults. For example, in a longitudinal study of British adults aged 50 years or older, individuals who were current smokers had higher odds of pain at baseline but there was no evidence for increased likelihood of new-onset widespread pain over 3 years of follow-up when former and current smokers were combined in one group and compared with never smokers.[23] Likewise, a cross-sectional study found that smoking status was only associated with chronic pain prevalence when current and former smokers were combined in one group and compared with never smokers among people aged 65 years and older in Sweden.[26] By using data collected prospectively across adulthood, we were able to distinguish different lifetime patterns of cigarette smoking and go beyond previous research. This has strengths when compared with other studies that have relied on single assessments of smoking and may therefore not adequately or reliably capture all relevant lifetime exposure.

### Potential explanations of findings

Our finding of increased risk of CWP among all smoking groups (ie, predominantly non-smokers, predominantly smokers and lifelong smokers), when compared with never smokers suggests that cigarette smoking even for a short period of time over the life course may have a long-lasting impact on the probability of reporting CWP in later adulthood. One plausible explanation for this finding is that smoking may produce changes in central nervous system function that increase susceptibility to pain the effects of which continue to persist after smoking cessation.[27] Another consideration is that the associations between smoking and pain may be bi-directional; not only is there evidence that tobacco smoking may be a risk factor for the development and exacerbation of pain but also that pain may serve to motivate smoking.[7] For this reason, it is possible that some individuals in our study may have developed CWP in early adulthood and

**Table 2** Associations between lifetime smoking history and pain at age 68 (n=2347)

| | CWP versus no pain | | CRP versus no pain | | Other pain versus no pain | |
|---|---|---|---|---|---|---|
| | RRR (95% CI) | P values | RRR (95% CI) | P values | RRR (95% CI) | P values |
| **Model 1** | | | | | | |
| Never smoker | 1 | 0.0001 | 1 | 0.25 | 1 | 0.25 |
| Predominantly non-smoker | 1.80 (1.26 to 2.58) | | 1.08 (0.86 to 1.35) | | 1.14 (0.85 to 1.52) | |
| Predominantly smoker | 2.64 (1.74 to 3.99) | | 1.32 (0.99 to 1.74) | | 1.44 (1.01 to 2.04) | |
| Lifelong smoker | 2.27 (1.25 to 4.13) | | 0.97 (0.62 to 1.51) | | 1.05 (0.60 to 1.84) | |
| **Model 2** | | | | | | |
| Never smoker | 1 | <0.001 | 1 | 0.33 | 1 | 0.09 |
| Predominantly non-smoker | 1.80 (1.26 to 2.59) | | 1.08 (0.86 to 1.35) | | 1.16 (0.87 to 1.55) | |
| Predominantly smoker | 2.52 (1.65 to 3.84) | | 1.29 (0.97 to 1.71) | | 1.51 (1.06 to 2.17) | |
| Lifelong smoker | 2.11 (1.15 to 3.86) | | 0.94 (0.60 to 1.47) | | 1.13 (0.64 to 1.99) | |
| **Model 3** | | | | | | |
| Never smoker | 1 | <0.001 | 1 | 0.46 | 1 | 0.17 |
| Predominantly non-smoker | 1.85 (1.28 to 2.66) | | 1.06 (0.84 to 1.33) | | 1.09 (0.82 to 1.47) | |
| Predominantly smoker | 2.43 (1.59 to 3.71) | | 1.22 (0.92 to 1.62) | | 1.37 (0.96 to 1.96) | |
| Lifelong smoker | 2.12 (1.15 to 3.91) | | 0.95 (0.60 to 1.48) | | 1.12 (0.63 to 1.99) | |
| **Model 4** | | | | | | |
| Never smoker | 1 | <0.001 | 1 | 0.45 | 1 | 0.20 |
| Predominantly non-smoker | 1.69 (1.17 to 2.45) | | 1.05 (0.83 to 1.31) | | 1.12 (0.84 to 1.50) | |
| Predominantly smoker | 2.33 (1.52 to 3.58) | | 1.23 (0.93 to 1.64) | | 1.40 (0.98 to 2.00) | |
| Lifelong smoker | 2.08 (1.12 to 3.86) | | 0.93 (0.59 to 1.46) | | 1.03 (0.59 to 1.81) | |
| **Model 5** | | | | | | |
| Never smoker | 1 | <0.001 | 1 | 0.37 | 1 | 0.20 |
| Predominantly non-smoker | 1.77 (1.23 to 2.55) | | 1.07 (0.86 to 1.34) | | 1.13 (0.85 to 1.51) | |
| Predominantly smoker | 2.47 (1.62 to 3.76) | | 1.27 (0.96 to 1.68) | | 1.40 (0.99 to 2.00) | |
| Lifelong smoker | 2.13 (1.16 to 3.91) | | 0.93 (0.60 to 1.46) | | 1.02 (0.58 to 1.80) | |
| **Model 6** | | | | | | |
| Never smoker | 1 | 0.002 | 1 | 0.90 | 1 | 0.15 |
| Predominantly non-smoker | 1.70 (1.16 to 2.49) | | 1.01 (0.80 to 1.28) | | 1.10 (0.82 to 1.48) | |
| Predominantly smoker | 2.10 (1.34 to 3.28) | | 1.12 (0.83 to 1.51) | | 1.39 (0.97 to 2.01) | |
| Lifelong smoker | 1.88 (0.99 to 3.57) | | 0.87 (0.54 to 1.38) | | 1.15 (0.64 to 2.06) | |

Model adjustments:
1: sex.
2: sex and socio-economic position (education at age 26, own occupational class).
3: sex and other health behaviours (BMI, alcohol intake and physical activity).
4: sex and long-standing illness or health problems.
5: sex and symptoms of anxiety and depression (GHQ-28).
6: all covariates included in models 1 to 5.
BMI, body mass index; CRP, chronic regional pain; CWP, chronic widespread pain; GHQ-28, 28-item General Health Questionnaire; RRR, relative risk ratio.

used smoking as a coping strategy for this with this then causing them to experience greater difficulty quitting.[28]

Those associations were observed between lifetime smoking and CWP but not with CRP or other pain suggests that any influence of smoking on pain is likely to be operating via mechanisms related specifically to the development of CWP. This is consistent with the idea that CWP and CRP may have different underlying aetiologies.[7] For example, some studies have suggested

that nicotine may alter pain regulatory mechanisms (eg, potential alteration of pain processing or cause structural changes to other systems) which might have long-term consequences for painful chronic conditions.[29] The associations between lifetime smoking and CWP were only partially attenuated after adjustment for a range of factors including sex, socioeconomic position, body size, other health behaviour, long-standing illness and symptoms of anxiety and depression. This suggests that

there might be other relevant explanatory variables. For example, although affective symptoms have been taken into consideration in our study some psychosocial models have highlighted the role of pain-related stress, fear or anxiety sensitivity in the transition from acute to chronic pain.[7 30 31] These pain-specific measures were not available in our data, and therefore, future studies are encouraged to include these and examine their association with CWP onset and smoking.

### Methodological considerations

One of the main strengths of our study was the use of data from a large nationally representative sample of older adults with prospective ascertainment of smoking status across adulthood which allowed us to characterise different lifetime patterns of smoking. Although we were unable to validate self-reports of smoking using cotinine measurements or reliably distinguish between light and heavy smokers, our approach does still have important strengths when compared with the commonly used categorisation of never, ex or current smoker based on self-reported smoking at a single time point. While the use of recalled information on quitting attempts between assessments could have introduced bias, when we performed sensitivity analyses to check for potential inconsistencies in reporting and compared results with retrospective measures on time since quitting for former smokers there were no differences in findings. However, the observed increase in prevalence of never smokers at older ages (online supplementary table 2) suggests that smokers were more likely than non-smokers to be lost to follow-up. This is partly explained by the fact that lifelong smokers had much greater rates of all-cause mortality across adulthood up to age 69 (online supplementary table 3); survival bias or differences in statistical power may therefore explain why predominantly smokers were found to have higher relative risks of CWP than lifelong smokers when both groups were compared with never smokers. However, another potential explanation of this finding which needs to be considered is the possibility that lifelong smokers may be less likely to experience CWP than predominantly smokers due to psychological or physiological effects.

Another potential limitation of our analyses is that we cannot rule out reverse causation as chronic pain was only assessed at one time point. As noted above, recent research has highlighted that the association between smoking and pain is likely to be bidirectional[28 32] and so future longitudinal studies with prospective ascertainment of both smoking and pain across life should be performed. Such studies would also benefit from assessment of pain severity and information on potential causes of pain, such as history of accident, which we were not able to take account of in our analyses but which may provide important additional insights. Although our findings are generalisable to the current generation of adults reaching old age in the UK, changing trends in smoking behaviour over the last 70 years[11 33] should be considered

and further research is needed to examine whether similar associations will be observed in more recently born cohorts who will have experienced different selection effects into smoking and different lifetime patterns of exposure to smoking.

## CONCLUSIONS

Our results highlight the role of smoking at any time point across adulthood as an independent risk factor for CWP in later adulthood, after considering socioeconomic, psychosocial and health-related factors. The evidence provided for the long-lasting effect of smoking on this condition highlights the potential benefit of considering smoking history rather than current smoking status when identifying those at greater risk of CWP in later life especially as opportunities to prevent CWP may be missed if only current smokers are classified as higher risk. These findings are relevant from a clinical and public health perspective, as smoking is a modifiable risk factor and our findings suggest that its primary prevention may be key.

**Acknowledgements** The authors are grateful to NSHD study members for their continuing support. We also thank members of the NSHD scientific and data collection teams who have been involved in NSHD data collections.

**Contributors** RB and SGM conceived the idea for this study and along with RC developed the study objectives. RC was part of the team responsible for the 68–69 years data collection. RB and SGM undertook statistical analyses. All authors contributed to the interpretation of the data. RB and SGM drafted the article and all authors contributed to its critical revision and provided final approval of the version to be published. RB and SGM are the guarantors and accept full responsibility for the work and the conduct of the study, had access to the data, and controlled the decision to publish. All authors had full access to all of the data (including statistical reports and tables) in the study and can take responsibility for the integrity of the data and the accuracy of the data analysis.

**Funding** This work was supported by the UK Medical Research Council who provide support for RC and SGM (Programme code MC_UU_12019/4) and fund the MRC National Survey of Health and Development. RB was supported in part by grant MR/R016372/1 for the King's College London MRC Skills Development Fellowship programme funded by the UK Medical Research Council (MRC) and by grant IS-BRC-1215-20018 for the National Institute for Health Research (NIHR) Biomedical Research Centre at South London and Maudsley NHS Foundation Trust and King's College London.

**Disclaimer** The views expressed are those of the authors and not necessarily those of the MRC, NHS, the NIHR or the Department of Health and Social Care. The funders of the study had no role in the study design, data collection, data analysis, data interpretation, writing of the report or the decision to submit the article for publication.

**Competing interests** RC, RB and SGM received financial support from the UK Medical Research Council for the submitted work.

**Patient consent** Obtained.

**Provenance and peer review** Not commissioned; externally peer reviewed.

**Data sharing statement** Data are available on request to the NSHD Data Sharing Committee. NSHD data sharing policies and processes meet the requirements and expectations of MRC policy on sharing of data from population and patient cohorts: http://www.mrc.ac.uk/research/research-policy-ethics/data-sharing/policy/. Data requests should be submitted to mrclha.swiftinfo@ucl.ac.uk; further details can be found at http://www.nshd.mrc.ac.uk/data.aspx DOI: 10.5522/NSHD/Q101;10.5522/NSHD/Q102; 10.5522/NSHD/Q103. These policies and processes are in place to ensure that the use of data from this 71-year-old national birth cohort study is within the bounds of consent given previously by study members, complies with MRC guidance on ethics and research governance, and meets rigorous MRC data security standards.

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
