## [Reviewer comments · BMJ Open]

ARTICLE DETAILS

TITLE (PROVISIONAL)	Lifetime cigarette smoking and chronic widespread and regional pain in later adulthood: Evidence from the 1946 British birth cohort study
AUTHORS	Bendayan, Rebecca; Cooper, Rachel; Muthuri, Stella

VERSION 1 – REVIEW

REVIEWER	Lori Bastian MD Yale University and VA Connecticut West Haven, CT, US
REVIEW RETURNED	12-Mar-2018

GENERAL COMMENTS	The study examined lifetime cigarette smoking exposure and pain in a 1946 British birth cohort. Participants included over 2000 men and women who completed a survey about their pain at age =68 and had been followed over time (smoking status was asked at 8 time points since birth). Overall, predominantly smokers, predominantly non-smokers, and lifelong smokers had an increased risk of chronic widespread pain (but not chronic regional pain) compared to never smokers. This manuscript is generally well-written. Comments and areas of suggested revision are noted: ABSTRACT: Add how smoking groups were defined INTRODUCTION: Page 4, 1st para, references 5 and 6 are about physical activity and BMI and pain and do not report on tobacco use. Authors note weak association of smoking and pain but do not cite studies showing a stronger association such as John et al Preventive Medicine 2006;43:477-81 and Volkman et al. Pain Medicine 2015;16:1690-6. METHODS: Page 6, Assessment of pain, the primary outcome, does not include any measure of severity/intensity of pain. A person with mild pain for >3 months is combined with someone who has severe and disabling pain. I wonder if the widespread pain (compared to regional pain) serves as a marker for severity. This is a significant limitation. Page 6, Assessment of smoking does not take into account the number of cigarettes smoked. Light and heavy smokers are combined and this may explain why the authors were not able to demonstrate a dose-response relationship. DISCUSSION: Page 12, paragraph comparing results to other studies, notes that the study by McBeth et al (ref 23) did not show former or current smokers had an increased likelihood of new onset pain compared to never smokers. McBeth did show that current smokers had higher odds of pain at baseline and then they combined former and current in their longitudinal analyses and this was not significantly associated with new pain. The authors also note that the Study by Jakobsson (ref 26) did not show and association between smoking and pain in an older cohort. My read of ref 26 is that there was an association between smoking and
---

	chronic pain among older people, especially regarding pain intensity. The authors should consider reframing the results from these other studies. The limitations section briefly mentions temporality that pain may have started at younger age and then led to smoking to cope with pain. Since the authors only have one measure of pain, they can't tell if the smoking preceded the pain. Overall, the conclusion highlights the importance of smoking primary prevention but these data do not suggest a role for smoking cessation programs.
--	---

REVIEWER	Jaana Keto University of Oulu, Finland
REVIEW RETURNED	18-Mar-2018

GENERAL COMMENTS	This is a well-written, easy-to-follow manuscript. I have a few suggestions for the authors. 1) The clinical implications of the results could be elaborated. Why should people who are at greater risk of CWP in later life because of their smoking history be identified? What actions should be taken? According to the results presented in the manuscript, smoking cessation isn't useful in preventing CWP. 2) In Results, the effect of different adjustments models (Table 2) could be phrased differently. I wouldn't say that the RRR for CWP vs. no pain between never-smokers and other groups were attenuated in Models 2, 3, 4, and 5 compared to Model 1. I wouldn't start describing the effects of Model 6 on the results with the word "however" either. I would highlight the fact that all the models (1-6) produced relatively similar results - this is an important finding. 3) In Conclusions, under Methodological considerations, the authors may want to discuss bioassays such as cotinine measurements (why weren't they used to confirm smoking status, what is the reliability of smoking status used in this study etc.) 4) Participation of leisure time physical activity is phrased differently in Methods and Table 1 (e.g. "more active" vs. "regularly active"). The authors may want to check this.
--

REVIEWER	Fawaz Mzayek School of Public Health, University of Memphis. USA
REVIEW RETURNED	21-Mar-2018

GENERAL COMMENTS	Re: Manuscript titled "Lifetime cigarette smoking and chronic widespread and regional pain in later adulthood: Evidence from the 1946 British birth cohort study" The study examined the potential association of life-long smoking patterns with diffuse chronic pain in a birth cohort that was established in 1946 with data available until age of 68. The manuscript is well written and the methodology is adequate. The authors are commended for their awareness and consideration of the potential of survival bias and reverse causality. However, few points can be addressed to improve the study. Methods:
--

	1- The outcome, chronic widespread pain, was defined according to criteria used for fibromyalgia, which is a painful, mostly diffuse, condition in itself. I am not sure how this definition is applicable to general body aching that usually comes with aging. 2- One important factor that must be taken into account, is the intensity of smoking, not only the duration. Number of cigarette per day should be available—since information of smoking seems to have been collected systematically. If this is true, it should be factored in the analysis. 3- A history of having an accident could potentially be an important confounder of the studied association. If this information is available, adjusting for it will greatly improve the study. Results: 1- In Table 2, the “Predominantly smoker” group exhibited stronger associations than the “Lifelong smoker” across all outcomes and models. It will be interesting to hear the opinion of the researchers on this observation, given their knowledge of the data. 2- In supplemental Table 1, the “never-smoker” proportion decreases from age 20 to 36, as is expected, and then it starts to increase. This could be a manifestation of survival bias, or differential loss to follow up among smokers. Discussing this observation will be a nice addition to the manuscript. Minor points: 1- Although occupation was adjusted for, but this was done as an indicator of SES. It would be interesting if occupation is analyzed as manual vs. non-manual. Because manual work could be a strong confounder (manual-type of work is associated with more smoking and more chronic pain later in life.) 2- Page 7, delete “and mediators”—none of the mentioned variables can be considered a mediator (i.e., on the causal pathway from the exposure to the outcome) 3- Page 11, lines 45-52, this notion is not clear in the context of the results. If nicotine diminishes pain sensitivity, then smokers should experience less pain.
--	--

VERSION 1 – AUTHOR RESPONSE

Reviewer: 1

The study examined lifetime cigarette smoking exposure and pain in a 1946 British birth cohort. Participants included over 2000 men and women who completed a survey about their pain at age =68 and had been followed over time (smoking status was asked at 8 time points since birth). Overall, predominantly smokers, predominantly non-smokers, and lifelong smokers had an increased risk of chronic widespread pain (but not chronic regional pain) compared to never smokers. This manuscript is generally well-written.

Response: We thank the reviewer for their positive assessment of our manuscript.

ABSTRACT: Add how smoking groups were defined

Response: We have revised the abstract so that the four different lifetime patterns of smoking are now described in the section on participants. Unfortunately it is not possible for us to provide more detailed definitions without exceeding the abstract word limit.

INTRODUCTION: Page 4, 1st para, references 5 and 6 are about physical activity and BMI and pain and do not report on tobacco use. Authors note weak association of smoking and pain but do not cite studies showing a stronger association such as John et al Preventive Medicine 2006;43:477-81 and Volkman et al. Pain Medicine 2015;16:1690-6.

Response: We apologise for this error; we have removed the original references 5 and 6 which had been cited by mistake and have now updated the references cited to include the reviews on smoking and pain by Ditre et al. (2011) and Parkerson et al. (2013). We thank the reviewer for highlighting the papers from John et al and Volkman et al. which we have now cited (references 9 and 10).

METHODS: Page 6, Assessment of pain, the primary outcome, does not include any measure of severity/intensity of pain. A person with mild pain for >3 months is combined with someone who has severe and disabling pain. I wonder if the widespread pain (compared to regional pain) serves as a marker for severity. This is a significant limitation.

Response: We agree with the reviewer that this is a limitation of our analyses and so this is now acknowledged in the discussion, page 13.

Page 6, Assessment of smoking does not take into account the number of cigarettes smoked. Light and heavy smokers are combined and this may explain why the authors were not able to demonstrate a dose-response relationship.

Response: We recognise that this is another limitation of our study and so this is now acknowledged in the discussion, pages 12-13. While we were unable to distinguish between light and heavy smokers as data on number of cigarettes smoked were judged to be less reliable and were more often missing, our approach does still have important strengths when compared with the more traditional approach of assessing smoking at a single time point and categorising this as never, ex or current smoker. That we did not demonstrate a dose-response relationship is more likely to be explained by the under-representation of lifelong smokers in our analytical sample; as we report in the discussion and present in Supplementary Table 3, these participants had much greater rates of all-cause mortality across adulthood.

DISCUSSION: Page 12, paragraph comparing results to other studies, notes that the study by McBeth et al (ref 23) did not show former or current smokers had an increased likelihood of new onset pain compared to never smokers. McBeth did show that current smokers had higher odds of pain at baseline and then they combined former and current in their longitudinal analyses and this was not significantly associated with new pain. The authors also note that the Study by Jakobsson (ref 26) did not show an association between smoking and pain in an older cohort. My read of ref 26 is that there was an association between smoking and chronic pain among older people, especially regarding pain intensity. The authors should consider reframing the results from these other studies.

Response: We agree with the reviewer that McBeth's findings were not clear and so we have now clarified this, page 11. As the reviewer notes, Jakobsson et al did find an association but only with chronic pain prevalence when current and former smokers were combined in one group and compared to never-smokers. We have now also clarified this, page 11.

The limitations section briefly mentions temporality that pain may have started at younger age and then led to smoking to cope with pain. Since the authors only have one measure of pain, they can't tell if the smoking preceded the pain. Overall, the conclusion highlights the importance of smoking primary prevention but these data do not suggest a role for smoking cessation programs.

Response: We agree with the reviewer; as we had outlined in the discussion 'another potential limitation of our analyses is that we cannot rule out reverse causation'. We also highlight this in the 'strength and limitations of this study' section. In light of this comment, we have removed reference to smoking intervention initiatives in the conclusions section of the discussion.

Reviewer: 2

This is a well-written, easy-to-follow manuscript. I have a few suggestions for the authors.

Response: We thank the reviewer for their positive assessment of our manuscript.

1) The clinical implications of the results could be elaborated. Why should people who are at greater risk of CWP in later life because of their smoking history be identified? What actions should be taken? According to the results presented in the manuscript, smoking cessation isn't useful in preventing CWP.

Response: We have expanded our discussion on the implications of the results, page 14. This includes highlighting that opportunities to prevent CWP may be missed if only current smokers are classified as higher risk and that our findings suggest that primary prevention of smoking is key.

2) In Results, the effect of different adjustments models (Table 2) could be phrased differently. I wouldn't say that the RRR for CWP vs. no pain between never-smokers and other groups were attenuated in Models 2, 3, 4, and 5 compared to Model 1. I wouldn't start describing the effects of Model 6 on the results with the word "however" either. I would highlight the fact that all the models (1-6) produced relatively similar results - this is an important finding.

Response: Thank you for this suggestion we have revised the relevant paragraph of the results section to make this clearer, page 9.

3) In Conclusions, under Methodological considerations, the authors may want to discuss bioassays such as cotinine measurements (why weren't they used to confirm smoking status, what is the reliability of smoking status used in this study etc.)

Response: We now acknowledge that a limitation of our study is that we were unable to validate reports of smoking using cotinine measurements, page 12. However, in other studies with relevant data, self-reports of smoking have been found to be reliable - see for example: Vartiainen, E., Seppälä, T., Lillsunde, P., & Puska, P. (2002). Validation of self-reported smoking by serum cotinine measurement in a community-based study. *Journal of Epidemiology & Community Health*, 56(3), 167-170).

4) Participation of leisure time physical activity is phrased differently in Methods and Table 1 (e.g. "more active" vs. "regularly active"). The authors may want to check this.

Response: We apologise for this inconsistency in how the categories had been referred to which has now been corrected so that the same terms are used to describe these categories in the text and tables.

Reviewer: 3

The study examined the potential association of life-long smoking patterns with diffuse chronic pain in a birth cohort that was established in 1946 with data available until age of 68. The manuscript is well written and the methodology is adequate. The authors are commended for their awareness and consideration of the potential of survival bias and reverse causality. However, few points can be addressed to improve the study.

Response: We thank the reviewer for their positive assessment of our manuscript.

Methods:

1- The outcome, chronic widespread pain, was defined according to criteria used for fibromyalgia, which is a painful, mostly diffuse, condition in itself. I am not sure how this definition is applicable to general body aching that usually comes with aging.

Response: A strength of our analyses is that we were able to use an outcome defined according to American College of Rheumatology criteria. Unfortunately, we do not have data on underlying causes of pain to allow us to distinguish specific clinical conditions from general body aching that may be related to ageing. Related to this is the fact that we were unable to consider pain severity which is now acknowledged as a limitation in the discussion, page 13.

2- One important factor that must be taken into account, is the intensity of smoking, not only the duration. Number of cigarette per day should be available—since information of smoking seems to have been collected systematically. If this is true, it should be factored in the analysis.

Response: As noted in our response to reviewer 1, see above, a limitation of our approach is that we could not take account of intensity of smoking and so this is now acknowledged in the discussion, pages 12-13. While we were unable to take account of smoking intensity as data on number of cigarettes smoked were judged to be less reliable and were more often missing, our approach does still have important strengths when compared with the more traditional approach of assessing smoking at a single time point and categorising this as never, ex or current smoker.

3- A history of having an accident could potentially be an important confounder of the studied association. If this information is available, adjusting for it will greatly improve the study.

Response: This is a very good suggestion but unfortunately there are no data on history of accident in NSHD so we cannot take account of this in our analyses. This is now acknowledged as a potential limitation, page 13.

Results:

1- In Table 2, the “Predominantly smoker” group exhibited stronger associations than the “Lifelong smoker” across all outcomes and models. It will be interesting to hear the opinion of the researchers on this observation, given their knowledge of the data.

Response: We think that this is most likely to be explained by the fact that lifelong smokers had greater rates of all-cause mortality and so were under-represented in our study sample. This is now suggested on page 12.

2- In supplemental Table 1, the “never-smoker” proportion decreases from age 20 to 36, as is expected, and then it starts to increase. This could be a manifestation of survival bias, or differential loss to follow up among smokers. Discussing this observation will be a nice addition to the manuscript.

Response: As the reviewer suggests, there are likely to be two main explanations for this change in prevalence of never smokers with age; survival bias and differential loss to follow-up (not due to death). This is now discussed on page 12.

Minor points:

1- Although occupation was adjusted for, but this was done as an indicator of SES. It would be interesting if occupation is analyzed as manual vs. non-manual. Because manual work could be a strong confounder (manual-type of work is associated with more smoking and more chronic pain later in life.)

Response: As the reviewer has noted, there are two benefits of adjusting for occupational class. Firstly, it is a useful indicator of socioeconomic position and secondly, it allows us to take account of manual work. This is now acknowledged in the methods, page 8. The 3 category grouping of occupational class used in our analyses makes a distinction between manual and non-manual work and this is now clarified by including more details of the types of work captured by each category on page 8. We prefer to use a 3 category variable rather than the 2 category comparison suggested (i.e. manual vs non-manual) as there are marked differences between those with an RGSC of I/II/M when compared with those with an RGSC of IV or V whereby if we were to combine these groups, residual confounding due to heterogeneity within categories would become a concern.

2- Page 7, delete “and mediators”—none of the mentioned variables can be considered a mediator (i.e., on the causal pathway from the exposure to the outcome)

Response: We have deleted ‘and mediators’ as requested.

3- Page 11, lines 45-52, this notion is not clear in the context of the results. If nicotine diminishes

pain sensitivity, then smokers should experience less pain.

Response: We agree that the example on pain sensitivity, which is derived from pain induction experimental studies, might not be clear without a more detailed explanation. Therefore, we have reworded it and now included the potential alteration of pain processing, page 12.

FORMATTING AMENDMENTS

Required amendments will be listed here; please include these changes in your revised version:

- Patient and Public Involvement statement:

We have implemented an additional requirement to all articles to include 'Patient and Public Involvement statement' within the main text of your main document. Please refer below for more information regarding this new instruction:

Authors must include a statement in the methods section of the manuscript under the sub-heading 'Patient and Public Involvement'.

This should provide a brief response to the following questions:

How was the development of the research question and outcome measures informed by patients' priorities, experience, and preferences?

How did you involve patients in the design of this study?

Were patients involved in the recruitment to and conduct of the study?

How will the results be disseminated to study participants?

For randomised controlled trials, was the burden of the intervention assessed by patients themselves?

Patient advisers should also be thanked in the contributorship statement/acknowledgements.

If patients and or public were not involved please state this.

Response: We have added a patient and public involvement statement to the methods section as requested, page 6.

VERSION 2 – REVIEW

REVIEWER	Lori Bastian MD, MPH Yale University and VA Connecticut Healthcare System, West Haven CT USA
REVIEW RETURNED	22-Apr-2018
GENERAL COMMENTS	Authors have done a complete job responding to my earlier concerns.
REVIEWER	Jaana Keto Centre for lifecourse health research, University of Oulu
REVIEW RETURNED	05-May-2018
GENERAL COMMENTS	As far as I can see, the authors were able to address concerns raised by the reviewers.
REVIEWER	Fawaz Mzayek, MD, PhD The University of Memphis, USA
REVIEW RETURNED	04-May-2018
GENERAL COMMENTS	The revised manuscripts is improved considerably. Two points need more clarification:

	1- The authors explained the observed stronger association of "predominantly smokers" with CWP, in comparison to "lifelong smokers", as a result of more censoring--potentially due to survival bias--in the latter group. While this could be a valid explanation, in the absence of evidence of differential censoring in terms of the outcome, more loss to follow up in the "lifelong smoker" group would affect the power of the analysis, rather than the magnitude of the effect. Another explanation would be that the lifelong smokers actually experience less CWP. This could be either a result of a psychological effect (smoking as a coping mechanism for pain is working), or a physiological effect (excess nicotine may alter pain sensitivity), as some evidence suggest. 2- The authors did not provide any information about the effects of the other variables that were adjusted for in the multivariable analysis. If some of these variables were significant, then it will be helpful to report that, otherwise a short statement that none was significant would suffice.
--	--

VERSION 2 – AUTHOR RESPONSE

Reviewer: 1

Reviewer Name: Lori Bastian MD, MPH

Authors have done a complete job responding to my earlier concerns.

Reviewer: 2

Reviewer Name: Jaana Keto

As far as I can see, the authors were able to address concerns raised by the reviewers.

Response: We thank reviewers 1 and 2 for taking the time to assess our paper again. We are pleased that they are satisfied with our revisions.

Reviewer: 3

Reviewer Name: Fawaz Mzayek, MD, PhD

The revised manuscripts is improved considerably.

Response: We thank reviewer 3 for assessing our paper again. We are pleased that they think our manuscript has been improved – this is testament to their previous comments.

Two points need more clarification:

1- The authors explained the observed stronger association of "predominantly smokers" with CWP, in comparison to "lifelong smokers", as a result of more censoring--potentially due to survival bias--in the latter group. While this could be a valid explanation, in the absence of evidence of differential censoring in terms of the outcome, more loss to follow up in the "lifelong smoker" group would affect the power of the analysis, rather than the magnitude of the effect. Another explanation would be that the lifelong smokers actually experience less CWP. This could be either a result of a psychological effect (smoking as a coping mechanism for pain is working), or a physiological effect (excess nicotine may alter pain sensitivity), as some evidence suggest.

Response: Thank you for this helpful suggestion. We have amended the discussion, page 13, to acknowledge these additional possible explanations of our findings.

2- The authors did not provide any information about the effects of the other variables that were adjusted for in the multivariable analysis. If some of these variables were significant, then it will be helpful to report that, otherwise a short statement that none was significant would suffice.

Response: We apologise for this omission. All covariates were associated with pain outcomes in the expected directions, hence their selection for inclusion in our models. This is now clarified in the results section on page 10.